# One-Pot Multicomponent Synthesis and Bioevaluation of Tetrahydroquinoline Derivatives as Potential Antioxidants, α-Amylase Enzyme Inhibitors, Anti-Cancerous and Anti-Inflammatory Agents

**DOI:** 10.3390/molecules25112710

**Published:** 2020-06-11

**Authors:** Samra Farooq, Aqsa Mazhar, Areej Ghouri, Naseem Ullah

**Affiliations:** Department of Pharmacy, Faculty of Biological Science, Quaid-I-Azam University, Islamabad 45320, Pakistan; samrafarooq@bs.qau.edu.pk (S.F.); aqsamazhar1947@gmail.com (A.M.); areejkhan27@gmail.com (A.G.); ihaq@qau.edu.pk (I.-U.-H.)

**Keywords:** tetrahydroquinoline, antioxidant, α-amylase enzyme inhibition, MTT assay, in vivo anti-inflammatory activity

## Abstract

Mankind has always suffered from multiple diseases. Therefore, there has been a rigorous need in the field of medicinal chemistry for the design and discovery of new and potent molecular entities. In this work, thirteen tetrahydroquinoline derivatives were synthesized and evaluated biologically for their antioxidant, α-amylase enzyme inhibitory, anti-proliferative and anti-inflammatory activities. **SF8** showed the lowest IC_50_ of 29.19 ± 0.25 µg/mL by scavenging DPPH free radicals. **SF5** showed significant antioxidant activity in total antioxidant capacity (TAC) and total reducing power (TRP) assays. **SF5** and **SF9** showed the maximum inhibition of α-amylase enzyme i.e., 97.47% and 89.93%, respectively, at 200 µg/mL concentration. Five compounds were shortlisted to determine their anti-proliferative potential against Hep-2C cells. The study was conducted for 24, 48 and 72 h. **SF8** showed significant results, having an IC_50_ value of 11.9 ± 1.04 µM at 72 h when compared with standard cisplatin (IC_50_ value of 14.6 ± 1.01 µM). An in vitro nitric oxide (NO) assay was performed to select compounds for in vivo anti-inflammatory activity evaluation. **SF13** scavenged the NO level to a maximum of 85% at 50 µM concentration, followed by **SF1** and **SF2**. Based on the NO scavenging assay results, in vivo anti-inflammatory studies were also performed and the results showed significant activity compared to the standard, acetylsalicylic acid (ASA).

## 1. Introduction

The field of drug discovery has conventionally been a dynamic, innovation-driven and highly prosperous sector of the global industry. Revolutionary advancements in health disciplines and technology have helped researchers in the discovery and development of numerous drugs, which play a significant role in the pharmacotherapy of a range of ailments and diseases [1]. Innovations in medicinal chemistry are responsible for the discovery of many breakthrough remedies that have ultimately improved the life and health of human beings over the past century. Continuous efforts in the development and evaluation of new molecules are required to drive the discovery of new medicines [2]. Numerous studies have been focused on the treatment of medical problems such as cancer, oxidative stress, inflammatory disorders, diabetes, etc.

The human body produces free radicals in the form of reactive oxygen species (ROS), which regulate cellular processes, i.e., angiogenesis, platelet aggregation, signal transduction, etc. When produced in excess, free radicals can impose a harmful and deleterious impact on cellular components, i.e., lipids, proteins, DNA, etc. This may lead to many chronic diseases, such as diabetes, cancer and atherosclerosis [3]. Antioxidants can scavenge these free radicals and helps to regulate the progression and prevention of carcinogenesis [4,5]. Cancer is an enigma that has threatened human lives for ages. It has a major impact on society across the globe as a leading cause of death worldwide. Subsequently, the amount of resources spent on its treatment is increasing daily. Therefore, enormous efforts have been invested in the research and development of anti-cancer products both in industry and academia [6].

Diabetes mellitus is one of the most common and emerging diseases in the world [7]. α-Amylase is a key enzyme responsible for the hydrolysis of carbohydrates. Inhibiting α-amylase is a successful approach to controlling postprandial blood sugar within the acceptable range, which is a key factor in controlling diabetes mellitus (DM) [8,9,10].

Inflammation is an indication of many pathological conditions, such as rheumatoid arthritis, osteoarthritis, Alzheimer’s disease, hepatitis, cancer, pulmonary fibrosis and obesity-related diseases [11,12]. Lipopolysaccharide (LPS), a bacterial endotoxin, plays an important role during inflammation by regulating a cascade of cellular responses [13]. Throughout the inflammatory process, pro-inflammatory mediators, cytokines, nitric oxide (NO) and prostaglandin E2 are produced by the inducible isoform of nitric oxide synthase (NOS) and cyclooxygenase-2 (COX-2) [14]. NO is a free radical synthesized from L-arginine and catalyzed by iNOS [15]. Additionally, iNOS is closely associated with a variety of pathological conditions [16] and can act as a cytotoxic agent in pathological conditions [17]. Thus, the inhibition of the production of NO by iNOS can be an effective approach to treat and prevent inflammatory diseases [18].

The concept of multicomponent organic synthesis has received increased attention in organic chemistry since it provides access to diverse libraries of organic molecules [19]. Thus, multicomponent reactions have gained great importance, especially in medicinal chemistry, and are used for the preparation of novel therapeutic agents that have numerous applications [20]. Many biologically active compounds are obtained by using the multicomponent reaction (MCR) approach. The corresponding biological activities exhibited by organic molecules include antileishmanial [21], anti-inflammatory [22], ROCK inhibitors [23], neuroprotective agents [24], acetylcholinesterase inhibitors [25], antimicrobial [26,27,28,29,30], antioxidants [31], anti-tumor [32,33], anti-cancer [34,35,36,37,38,39] and so on.

Quinoline [40] (1-azanaphthalene) is a very significant nitrogen-containing heterocyclic ring structure in medicinal chemistry [41,42]. It has been integrated into many molecules, resulting in compounds with a broad spectrum of potential [43]. A literature survey revealed that quinoline derivatives possess a wide range of biological activities, such as anti-inflammatory, anti-malarial, anti-tumor, anti-hypertensive, anti-microbial, tyrosine PDGF-RTK inhibiting agents, anti-HIV and anti-tubercular, etc. [44,45,46]. Some of the promising compounds with a quinoline ring system are depicted in the Figure 1.

This diversity of biological effects is exhibited by benzofused six-membered heterocyclic rings. From a chemical point of view, the tetrahydroquinoline (THQ) moiety is a promising derivative of quinoline and a potential scaffold in medicinal chemistry due to its broad-spectrum biological profile, as mentioned in Figure 2. Helquinoline is a potent tetrahydroquinoline-derived antibiotic that has been extracted from *Janibecter limosus* [47]. In particular, tetrahydroquinoline exhibits a wide range of biological activities, for example, anti-HIV [48], antitrypanosomal [49], psychotropic [50], anti-inflammatory [51], antibacterial [52], antimalarial [53], antifungal [54] and antitumor activities [55].

The incorporation of a functional group into a pharmacophore is an attractive approach to the design and synthesis of new bioactive compounds. Considering the pharmacological potential of the tetrahydroquinoline moiety and our long-standing interest in exploring Mannich base chemistry [56], in the present study, we used a one-pot MCR approach to prepare a series of tetrahydroquinoline derivatives. The majority of the synthesized Mannich bases have not been reported in the literature to date. All compounds were subsequently evaluated in biological studies for their pharmacokinetic, antioxidant, α-amylase enzyme inhibition, cytotoxic and anti-inflammatory potential. The comprehensive biological profile of tetrahydroquinoline-derived N-Mannich bases by one-pot MCR reaction has not yet been thoroughly investigated. A literature survey demonstrated that there are no studies concerning the cytotoxic and anti-proliferative activities of tetrahydroquinoline-derived N-Mannich bases.

## 2. Results and Discussions

### 2.1. Chemistry

Multicomponent reactions (MCRs) are considered to be very important and powerful reactions in combinatorial and medicinal chemistry [57] because of the benefits they offer in terms of the synthesis of diverse structures that will lead to an increase in the economy of organic synthesis [58,59] by shortening the time span [60].

The reaction between aldehydes, amines or ammonia and heterocyclic acidic proton-containing pharmacophores is known as a Mannich reaction. It was reported for the first time by a German chemist, Carl Mannich, in 1912, for the synthesis of atropine glucoside [61,62].

The synthetic route used to obtain compounds (**SF1**–**SF13**) is described in Scheme 1. N-Mannich products were prepared by a one-pot, three-component condensation reaction, performed under reflux conditions. Formaldehyde (**S**), amine (**1**–**13**), and tetrahydroquinoline (**F**) were placed in a pressure tube and dissolved in ethanol and a catalytic amount of HCl was added. The mixture was refluxed at 80 °C for ~5–7 h, leading to the formation of the N-Mannich base. The product precipitated from the reaction mixture and the formed solid was separated, washed and recrystallized by ethanol. For some of the products, an additional purification step by flash column chromatography was necessary. To establish the exact position of the substituents of the titled compound, we hypothesized that the reaction was carried out at position 1, where the acidic proton-H of the parent compound reacts with the oxygen of formaldehyde (Formalin 37%) and takes one -H from the amines, yielding a water molecule as a byproduct during the reflux condensation reaction.

The Mannich reaction is a condensation reaction where the substrate used is XH compounds, with X being any heteroatom (C, N, S, etc.) with nucleophilic properties. The reaction is the nucleophilic addition of an amine to a carbonyl group of an aldehyde, followed by dehydration to a Schiff base, which acts as an electrophile and reacts with a compound containing an acidic proton. The reaction is an example of an S_N_2 addition reaction [56]. From the structure of tetrahydroquinoline, it should be kept in mind that it contains one nucleophilic site, i.e., −NH. Under a basic environment, the nucleophile site gets deprotonated, which results in an increase in electronic condensation on the electronegative nitrogen atom (nucleophilic activation). Thus, the tetrahydroquinoline anion starts a nucleophilic attack on the electrophile and carries out an efficient S_N_2 displacement and intramolecular cyclization. All the tetrahydroquinoline-derived Mannich bases were characterized by various spectroanalytical techniques. In the IR spectrum of quinoline derivatives, the strong bonds observed at around 1600 cm^−1^ and 1505 cm^−1^ correspond to aromatic C=C stretching. The specific carbonyl peak shown by some synthetic compounds appeared at 1620–1680 cm^−1^. In proton NMR, aromatic protons were observed mostly downfield, near δ 6.52–7.7 ppm, as per the substitution pattern at various positions of the aromatic ring. The methylene proton of −N linkage (N-CH_2_-N) was resonated at around δ 3.52–5.44 ppm and gave a singlet (s). Carbon NMR further confirmed the structure of the synthesized compounds. In carbon NMR of synthetic compounds, most downfield carbon was of aromatic quinoline carbon observed near δ 115–145 ppm. The carbon of methylene imine linkage was observed at δ ~69 ppm. Solvent peaks for CDCl_3_ and DMSO-*d*_6_ were observed at δ 77.2 ppm and 39.5 ppm, respectively. All other aromatic signals were observed at the expected regions. In the mass spectral analysis of synthetic compounds (**SF1**–**SF13**), the molecular ion peak for these compounds was observed as (M^+^) for all the compounds.

### 2.2. ADME Predictions

Synthetic compounds were evaluated for their in silico ADME for the assessment of pharmacokinetic parameters. A computational tool was used for the prediction of drug-likeness and molecular properties to obtain new drug-like leads. The values are presented in Table 1.

According to the biopharmaceutics drug disposition classification system (BDDCS), new molecules that are of large molecular weight, poorly water-soluble and lipophilic are categorized under BCS Class II. Lipinski et al. [63] demonstrated that molecules obtained via high-throughput screening (HTS) tend to have lipophilicity and greater molecular weights than leads in the pre-HTS era [64]. The Lipinski rule of five was developed to achieve the “druggability” of new molecular entities (NMEs) [65], which predicts that poor absorption or permeation is more likely when there are more than five H-bond donors, 10 H-bond acceptors, the molecular weight is greater than 500 and the calculated log *p* (CLog *p*) is greater than 5 [63,65]. In the current study, all the synthetic compounds were shown to have followed Lipinski’s rule of five, which demonstrates that these all fulfill the basic requirement for orally active drugs and tend to have low attrition rates during clinical trials.

### 2.3. Bioevaluation

#### 2.3.1. Antioxidant Assays

##### DPPH Assay

The % free radical scavenging activity (%FRSA) of the synthetic compounds was evaluated by the discoloration of an unstable DPPH reagent to the stable yellow-colored 2,2-diphenyl-1-picrylhydrazine molecule by accepting one electron from a donor antioxidant [66]; the mechanism is shown in Scheme 2.

The results of the %FRSA of the synthetic compounds and their IC_50_ data are summarized in Table 2.

All the synthetic compounds were shown to scavenge the DPPH free radical to some extent. **SF8**, which has an electronegative atom, was shown to have the most potent effect, with an IC_50_ value of 29.19 ± 0.25 µg/mL, which was lower than the standard, ascorbic acid and quercetin, with values of 41.38 ± 0.34 and 41.64 ± 1.01 µg/mL, respectively. Among the other synthetic series, **SF4**, **SF6** and **SF12** exhibited good antioxidant potential against DPPH free radicals, with IC_50_ values of 29.79 ± 0.26, 35.89 ± 0.33 and 39.33 ± 0.28 µg/mL, respectively. These characteristic results could be because of the formation of the hydrogen bond between an amine and a lone pair of nitrogen present on an unstable DPPH free radical. The high activity of aromatic amines is because of their ability to form nitroxyl radicals [68]. The literature survey supports the results of **SF6** and **SF8** being good antioxidants because of the nature of their substituents, i.e., electron-withdrawing (chlorine in **SF8**) and electron-donating (piperazine moiety in **SF6**) groups attached to the heterocyclic core [69].

##### Phosphomolybdenum-Based Total Antioxidant Capacity (TAC) and Total Reducing Power (TRP)

In the presence of an antioxidant, the phosphomolybdate ion gets reduced and a green phosphate–molybdate complex is formed, which is analyzed spectrophotometrically [70]. The highest TAC was found to be 285.46 ± 2.1 µg/AAE for **SF5** bearing a methyl piperazine moiety, followed by **SF7**–**SF9** and **SF12**, with TAC values of 242.77 ± 2.31 µg/AAE, 212.71 ± 231 µg/AAE, 179.76 ± 1.6231 µg/AAE and 191.19 ± 1.1231 µg/AAE, respectively (Figure 3 and Figure 4 (standard curve)).

A TRP assay was carried out to evaluate the reducing potential of synthetic compounds. The method implicates reductones. These reductones are species with antioxidant potential that is believed to be due to their proton-donating capacity [71]. This results in a discontinuation of a free radical chain. The maximum value of reducing power was 104.7 ± 1.9 µg/AAE for **SF5**, followed by 103.18 ± 2.4 µg/AAE for **SF7** and 84.57 ± 1.9 µg/AAE for **SF12** (Figure 3 and Figure 4 (standard curve)). The results indicate that synthetic compounds have the potential to stabilize free radicals by donating electrons and exhibited reductive potential with the highest reducing power [72]. The lowest reducing potential was observed for **SF1**, i.e., 33.22 ± 1.5 µg/AAE. **SF10** showed no activity in this assay. **SF7**, with a diphenylamine moiety, showed the highest activity in antioxidant assays due to the reason that secondary amines with an aromatic ring and nitrogen atom attached to them are good antioxidants [73] and are considered as singlet oxygen scavengers [74].

#### 2.3.2. α- Amylase Inhibitory Assay

In the present research, the inhibition of the α-amylase enzyme was assessed in vitro. The results are depicted in Figure 5.

Among the tested compounds, **SF5** showed the maximum % enzyme inhibition of 97.37%, followed by **SF9** and **SF7**, with 93%, and 88.67%, respectively. **SF2** showed a % enzyme inhibition close to the standard, acarbose, i.e., 79.97%. However, **SF13**, with an acetaminophen moiety, showed the lowest enzyme inhibition, i.e., 10.51%. The IC_50_ data were calculated by GraphPad Prism software. **SF5** showed the most potent value compared to the standard, acarbose (38.05 ± 0.98 µg/mL), followed by **SF9** and **SF8**, i.e., 18.55 ± 0.89 and 27.39 ± 1.43 µg/mL, respectively.

#### 2.3.3. Evaluation of Cytotoxic Potential

##### Brine Shrimp Cytotoxicity Assay

Cytotoxicity via a brine shrimp lethality assay was studied to reveal the cytotoxicity potential of the synthetic compounds. Toxicity to brine shrimps has a direct correlation with anti-cancer activity [75]. The brine shrimp larvae correspond to the mammalian system [76] since the DNA-dependent RNA polymerases of *Artemia salina* have been reported to be similar to the mammalian type [77]. This test is not only used to predict the anti-cancer potential of corresponding compounds but also anti-microbial and pesticidal behavior [78].

Thirteen of the newly synthesized compounds were screened against brine shrimp nauplii. The cytotoxicity of synthetic compounds was determined by calculating the median lethal concentration, LD_50_ (concentration at which 50% of the nauplii died). The results are shown in Table 3, indicating that **SF5**, **SF4**, **SF7** and **SF3** have good activity against brine shrimp nauplii, in comparison to a standard, doxorubicin (124 ± 1.54 µg/mL), with LD_50_ values of 94 ± 1.06, 100 ± 1.78, 100.8 ± 1.14 and 102 ± 1.98 µg/mL, respectively, which corresponds to a high toxicity against cancer cells. **SF1** and **SF8** showed moderate toxicity, 120.8 ± 2.3 µg/mL and 123.4 ± 1.87 µg/mL, respectively. The rest of the compounds exhibited weaker activity and were considered to be safe or non-toxic. Further cytotoxicity of the compounds was evaluated using cancer cell lines (Hep-2C).

##### Cytotoxicity Against Raw Macrophages and Cancer Cell Lines (Hep-2C)

Before the screening studies for the % cell viability inhibition against Hep-2C, it was essential to perform cytotoxicity tests against raw macrophages to determine the toxic and non-toxic behavior of every compound. The compounds that showed the lowest IC_50_ are considered to be toxic, as compared to the reference drug, and are screened out against cancer cells. The mechanism is depicted in Scheme 3.

The % cell viability of the synthetic library was evaluated in vitro by measuring the mitochondrial dehydrogenase activity (MTT test) on raw macrophages. All the compounds were shown to have a dose-dependent dose response (Table 4).

A 100 µM concentration was considered to be toxic for compounds because of the least number of viable cells. To determine the concentration required to achieve a 50% inhibition of cells induced by each compound, IC_50_ values (µM) were determined and their results were compared with a cytotoxic drug, i.e., doxorubicin. **SF1**, **SF4**, **SF5** and **SF9** exhibited the most potent activity, with IC_50_ values of 6.232 ± 0.01, 7.208 ± 0.05, 6.181 ± 0.05 and 14.24 ± 0.26 µM, respectively, while the standard drug doxorubicin had an IC_50_ value of 17.08 ± 0.37 µM, conclusively, which was considered toxic for cells. The activity of the most potent compounds, **SF1** and **SF5**, was three times more than that of doxorubicin. Piperidine-bearing compounds are known to have significant anti-cancer potential, as proved in the literature [79]. From the previous data, it is evident that the electronegativity of the substituents at the heterocyclic ring steer the magnitude of activity [80], as in the case of **SF9** with an electronegative bromine atom in it. The method is not specific to anti-tumor activity. However, a positive correlation was found between the least cell viability and the cytotoxicity toward some cancer cell lines [81].

##### MTT Assay Against Hep-2C Cells

From the results of the brine shrimp assay and cell cytotoxicity, compounds that showed significant activity and the lowest IC_50_ values in both assays were shortlisted for further analysis against Hep-2C cells. **SF1**, **SF4**, **SF5**, **SF7** and **SF8** showed cytotoxicity against brine shrimp nauplii at IC_50_ values of 120 ± 2.3, 100 ± 1.78, 94 ± 1.06, 100.8 ± 1.14 and 123.4 ± 1.87 µg/mL, respectively. The same five compounds showed good cytotoxic profiles against raw macrophages at IC_50_ values of 6.232 ± 0.01, 7.208 ± 0.05, 6.181 ± 0.05, 18.6 ± 0.13 and 21.23 ± 0.33 µM, respectively.

The results were plotted against % cell viability of Hep-2C cells and sample concentration, while cisplatin was taken as a standard (Figure 6). From the achieved results, it was clear that **SF1**, **SF4** and **SF7** exhibited better activity against the cell lines than **SF5** and **SF8**. All other compounds were not considered to have anti-cancer potential. The anti-cancer activity was determined at 24, 48 and 72 h (Figure 6). The anti-proliferative activity of Mannich bases is supposed to be due to the formation of enones [82].

IC_50_ values were determined by GraphPad software and plotted against time intervals (24 h, 48 h and 72 h) as summarized in Table 5. Previous data proved that heterocyclic compounds bearing piperidine in their structure are highly cytotoxic (**SF1**) [83]. Electronegative atoms impart moderate anti-tumor activity to compounds [16], as in **SF8**, which has a -chloro group in it. Electronegative or electron-withdrawing atoms will lead to an increase in cytotoxic potential (bromine, chlorine, fluorine, etc.). The structure–activity relationship (SAR) was established from these findings, which states that by derivatizing the already reported anti-cancerous lead molecule with secondary amines of cytotoxic potential will impart more cytotoxicity activity to the new product that has both nuclei. The observation of microscopic analysis of MTT stained cells and crystals is depicted in Figure 7.

#### 2.3.4. Anti-Inflammatory Activity

##### Nitric Oxide Scavenging Assay (NO)

Nitric oxide (NO) is an indicator of inflammation. It is an important chemical mediator produced by endothelial cells, macrophages and neurons, and is involved in the regulation of many different physiological processes [84]. An excess amount of nitric oxide (NO) is associated with several diseases, i.e., inflammation, cancer and other pathological conditions. Therefore, NO production inhibition is a principle key factor for the screening of drugs with anti-inflammatory potential [85].

A Griess reagent, a spectrophotometric determination of nitrite level, was carried out to measure the nitrite level in the conditioned medium of macrophages treated with lipopolysaccharide (LPS). Sodium nitrite (NaNO_2_) was used as a standard compound for the standard curve (Figure 8). The NO concentrations were obtained using the standard curve equation: y = 0.0083x + 0.0017, R^2^ = 0.9999, where y is the absorbance at 540 nm and x is the NO concentration in µM. The inhibitory activity of the tested compounds towards NO generated by LPS-induced macrophages was obtained from the calculated value of x.

The highest concentration to determine the inhibition of NO production was considered to be 100 µM, and from the results summarized in Table 6, it is evident that the synthetic compounds showed the concentration-dependent response. Among the thirteen compounds, the ones that inhibited the NO production to a greater extent even at 1 µM were considered to be potent and were considered for further in vivo assays.

The results from this finding showed that **SF13** has the highest NO inhibitory potential, with the maximum inhibition of 75% at 1 µM, followed by **SF1** and **SF2**, with NO production at 1 µM concentration of 27.771 ± 0.45 and 32.231 ± 0.46, respectively (Table 6).

The higher NO inhibition by the synthetic compounds corresponds to the epidemiological data that suggest that lower incidences of certain inflammatory diseases, i.e., cancer, arthritis, diabetes and acquired immune deficiency syndrome (AIDS) [86,87]. These results indicate that **SF13** and **SF1**–**SF3**, with acetaminophen, piperidine, morpholine and pyrrolidine amine incorporated into the basic tetrahydroquinoline ring, may suppress NO generation to a maximum extent by inhibiting iNOS enzyme activity. It is considered that the tested compounds that have shown the maximum decrease in NO production have beneficial therapeutic effects in the management of inflammatory conditions. Based on the results of NO, compounds were selected for in vivo anti-inflammatory activity.

##### In Vivo Anti-Inflammatory Activity

Synthetic compounds that have significantly inhibited the % nitric oxide (NO) production were further assessed for in vivo anti-inflammatory activity. Dose optimization results showed that doses of 0.1 and 1 mg/kg did not show any significant decrease in inflammatory behavior. However, at 10 mg/kg there was a significant decrease in paw edema (Figure 9). That is why the 10 mg/kg dose was selected for further study.

The inhibitory potential of **SF1**–**SF3** and **SF13** was evaluated by measuring paw thickness (Table 7). The synthetic compounds showed a significant decrease in paw edema with time duration as compared to the standard, acetylsalicylic acid (ASA). The results revealed that the compounds showed anti-inflammatory activity with meaningful statistical results. The results revealed that the compound with acetaminophen (**SF13**) significantly decreased the inflammation because of the acetaminophen moiety, which is categorized under non-steroidal anti-inflammatory drugs (NSAID) and has the potential to inhibit COX-1 and COX-2 enzymes involved in the production of prostaglandin [88]. Compounds **SF1**, **SF2** and **SF13**, with piperidine, morpholine and acetaminophen, also showed significant results in comparison to ASA.

#### 2.3.5. Structure-Activity Relationship (SAR)

The structure activity relationship is established from the findings of chemical moieties and their relationship to its biological activity. The results are summarized in Figure 10. 

## 3. Materials and Methods

Unless otherwise noted, all commercially available compounds and solvents were purchased from Sigma-Aldrich and Merck, Germany. Analytical thin-layer chromatography (TLC) was performed with aluminum sheets, silica gel 60 F254 (Merck), by a solvent system, EtOAc: pentane in a 7:2 ratio and the products were visualized with UV irradiation (254 nm). Gallenkamp melting point apparatus was used to determine the melting points with open capillaries that and are uncorrected. FTIR was recorded by the KBr pellets method by the PerkinElmer spectrum using the attenuated total reflectance (ATR). Nuclear magnetic resonance (NMR) spectra were recorded on an Agilent VNMR 400 (^1^H NMR: 400 MHz, ^13^C NMR: 101 MHz) or an Agilent VNMR 600 (^1^H NMR: 600 MHz, ^13^C NMR: 151 MHz) spectrometer. The chemical shifts are given in parts per million (ppm) relative to the residual solvent peak of the nondeuterated solvent (CDCl_3:_
^1^H NMR: δ = 7.26 ppm; ^13^C NMR: δ = 77.00 ppm). The multiplicity was reported with the following abbreviations: s = singlet, d = doublet, t = triplet, m = multiplet, *p* = pentet, br = broad signal, dd = doublet of doublet, dt = doublet of triplet, ddt = doublet of doublet of triplet, td = triplet of doublet, tp = triplet of pentet, tdd = triplet of doublet of doublet. Mass spectra were recorded on a Finnigan SSQ 7000 spectrometer. The PE 2400 Series II CHNS/O Analyzer was used to determine the content of carbon, hydrogen and nitrogen in organic materials.

### 3.1. General Procedure for the Synthesis of N-Mannich Bases of Tetrahydroquinoline

In a pressure tube equipped with a stirring bar, amines **1**–**13** (1.0 equiv, 0.5 mmol), formaldehyde **S** (2.0 equiv, 1 mmol) and tetrahydroquinoline **F** (1.0 equiv, 0.5 mmol) were added into 3 mL of ethanol and the mixture was heated to 80 °C in an oil bath under reflux conditions. Concentrated HCl (2 mol %) was added as a catalyst. The reaction progress and completion of the reaction was monitored by TLC. After ~5–7 h of refluxing, the reaction mixture was removed and cooled at room temperature. The reaction mixture was neutralized with a NaHCO_3_ solution and extracted with DCM (dichloromethane) (3 × 10 mL). The organic layer was dried over Na_2_SO_4_ and evaporated by a rotary evaporator. The product was washed and recrystallized by ethanol. For some of the compounds, an additional purification step of flash column chromatography was performed by using ethyl acetate: pentane in a 7:2 ratio.

The target compounds **SF1**–**SF13** were synthesized according to the synthetic route shown in Scheme 4.

*1- (Piperidine-1-ylmethyl) 1, 2, 3, 4-tetrahahydroquinoline* (**SF1**)

White solid; yield 82%; m.p. 130–137 °C; *R*_f_ = 0.89; IR (ATR) *υ_max_* 3397, 2850, 2327, 1710, 1613, 1437, 1191, 1042, 993 cm^−1^; ^1^H-NMR (600 MHz, CDCl_3_) δ ppm 6.94 (t, 1H), 6.91 (t, *J* = 4.7 Hz, 1H), 6.64 (dd, *J* = 7.0, 3.7 Hz, 1H), 6.60–6.54 (m, 1H), 4.70 (s, 2H), 3.27 (q, *J* = 5.5 Hz, 6H), 2.73 (t, 2H), 1.93 (td, *J* = 5.3, 2.6 Hz, 6H), 1.57 (t, 2H); ^13^C NMR (151 MHz, CDCl_3_) δ ppm 144.92, 127.11, 125.48, 63.50, 51.73, 47.07, 28.17, 25.92, 25.76, 22.35; EIMS *m*/*z* 230.2(M +), Anal. (C_15_H_22_N_2_): C 78.21, H 9.63, N 12.16, Calcd. C 78.27, H 9.61, N 12.19.

*4- ((3, 4-Dihydroquinolin—1(2H)-yl) methyl) morpholine* (**SF2**)

Light yellow solid; yield 84%; m.p. 142–148 °C; *R*_f_ = 0.63; IR (ATR) *υ_max_* 3390, 2834, 2322, 2083, 1898, 1663, 1611, 1503, 1304, 1049 cm^−1^; ^1^H NMR (600 MHz, CDCl_3_) δ ppm 7.04 (td, *J* = 7.9, 7.4, 1.6 Hz, 1H), 6.95 (dd, *J* = 7.4, 1.5 Hz, 1H), 6.88 (dd, *J* = 8.3, 1.1 Hz, 1H), 6.61 (td, *J* = 7.3, 1.1 Hz, 1H), 3.77 (s, 2H), 3.71 (t, 4H), 3.34 (t, 2H), 2.76 (t, *J* = 6.4 Hz, 2H), 2.50 (s, 4H), 1.93 (tt, 2H); ^13^C NMR (151 MHz, CDCl_3_) δ ppm 145.48, 129.03, 127.10, 126.88, 116.54, 67.01, 66.87, 52.00, 50.80, 27.94, 22.40; EIMS *m*/*z* 232 (M+), Anal. (C_14_H_20_N_2_O): C 72.38, H 8.68, N 12.06, Calcd: C 72.39, H 8.67, N 12.08.

*1-(Pyrollidin-1-ylmethyl) -1, 2, 3, 4-tetrahydroquinoline* (**SF3**)

Yellow solid; yield 80%; m.p. 115–117 °C; *R*_f_ = 0.86; IR (ATR) *υ_max_* 3388, 2921, 2323, 1990, 1656, 1501, 1300, 1153, 972 cm^−1^; ^1^H NMR (600 MHz, CDCl_3_) δ ppm 7.03 (td, 1H), 6.94 (dd, 1H), 6.86 (td, 1H), 6.60 (dd, 1H), 3.98 (s, 2H), 2.77 (t, *J* = 6.3 Hz, 2H), 2.60 (ddt, *J* = 6.7, 3.9, 2.1 Hz, 5H), 1.95 (t, 2H), 1.79 (ddd, *J* = 6.7, 4.6, 1.9 Hz, 5H); ^13^C NMR (151 MHz, CDCl_3_) δ ppm 145.48, 129.05, 126.90, 122.17, 116.16, 67.48, 51.84, 26.97, 23.53, 22.41; EIMS *m*/*z* 216 (M+), Anal. (C_14_H_20_N_2_): C 77.73, H 9.32, N 12.95, Calcd. C 77.77, H 9.31, N 12.97.

*N-((3, 4-dihydroquinolin-1(2H)-yl), methyl)-N-propylpropan-1 amine* (**SF4**)

Yellow powder; yield 79%; m.p. 123–132 °C; *R*_f_ = 0.74; IR (ATR) *υ_max_* 3841, 2831, 1685, 1510, 1304, 972 cm^−1^.; ^1^H NMR (600 MHz, CDCl_3_) δ ppm 6.98 (dd, 1H), 6.91 (td, 1H), 6.74 (dd, 1H), 6.66 (td, 1H), 4.71 (s, 2H), 3.29 (t, *J* = 11.4 Hz, 2H), 2.78–2.60 (m, 4H), 2.59 (t, 2H), 1.99–1.95 (m, 4H), 1.50 (t, 2H), 0.89 (tt, 6H); ^13^C NMR (151 MHz, CDCl_3_) δ ppm 145.95, 128.98, 127.17, 126.79, 122.44, 115.97, 67.48, 59.85, 46.94, 27.87, 22.41, 20.41, 11.82; EIMS *m*/*z* 246.22 (M+); Anal. (C_16_H_26_N_2_): C 77.99, H 10.64, N 11.37, Found: C 77.95, H 10.67, N 11.39.

*1-(4-Methylpiperazin-1-yl) methyl)-1, 2, 3, 4 -tetrahydroquinoline* (**SF5**)

Light yellow solid; yield 81%; m.p. 167 °C; *R*_f_ = 0.76; IR (ATR) *υ_max_* 3354, 3061, 1605, 1461, 1354, 1003, 949 cm^−1^; ^1^H NMR (600 MHz, CDCl_3_) δ ppm 7.01–6.99 (m, 1H), 6.88 (td, *J* = 7.5, 2.0 Hz, 1H), 6.56 (td, *J* = 7.5, 2.0 Hz, 1H), 6.37 (dd, *J* = 7.5, 2.0 Hz, 1H), 4.42 (s, 2H), 3.33 (t, *J* = 5.1 Hz, 2H), 2.90 (td, *J* = 6.2, 1.0 Hz, 2H), 2.51–2.47 (m, 9H), 2.24 (s, 3H), 2.08–2.04 (m, 2H); ^13^C NMR (151 MHz, CDCl_3_) δ ppm 145.62, 128.98, 126.89, 122.28, 116.37, 67.47, 54.95, 50.29, 48.41, 46.08, 27.91, 22.42; EIMS *m*/*z* 245.20 (M+); Anal. (C_15_H_23_N_3_): C 73.47, H 9.42, N 17.17, Found: C 73.43, H 9.45, N 17.13.

*1-(Piperazine-1-yl methyl)-1, 2, 3, 4-tetrahydroquinoline* (**SF6**)

Off-white crystals; yield 87%; m.p. 220–227 °C; *R*_f_ = 0.77; IR (ATR) *υ_max_* 3863, 3379, 2833, 2493, 2285, 2088, 1900, 1655, 1611, 1500, 1314, 1050 cm^−1^; ^1^H NMR (600 MHz, CDCl_3_) δ ppm 6.97–6.94 (m, 2H), 6.59 (dd, *J* = 7.4, 1.2 Hz, 1H), 6.47 (d, *J* = 1.1 Hz, 1H), 3.81 (s, 2H), 3.30–3.29 (m, 4H), 2.94–2.85 (m, 4H), 2.76 (t, *J* = 6.4 Hz, 4H), 2.17 (s, 1H), 1.95–1.93 (m, 2H); ^13^C NMR (151 MHz, CDCl_3_) δ ppm 145.48, 129.03, 127.10, 126.88, 122.36, 116.54, 67.01, 52.00, 50.80, 46.93, 28.09, 22.21; EIMS *m*/*z* 231.34 (M+). Anal. (C_14_H_21_N_3_): C 72.69, H 9.15, N 18.16, Found: C 72.66, H 9.19, N 18.21.

*N-((3, 4-dihydroquinolin-1(2H)-yl) methyl)-N-phenylaniline* (**SF7**)

Brown crystals; yield 82%; m.p. 243–248 °C; *R*_f_ = 0.81; IR (ATR) *υ_max_* 3378, 2921, 2692, 2494, 2285, 2110, 1904, 1794, 1592, 1311, 1174, 946 cm^−1^; ^1^H NMR (600 MHz, DMSO-*d_6_*) δ ppm) 7.24–7.20 (m, 8H), 7.05–6.99 (m, 3H), 6.88 (td, *J* = 7.5, 2.0 Hz, 1H), 6.56 (td, *J* = 7.5, 2.0 Hz, 1H), 6.38 (dd, *J* = 7.5, 2.0 Hz, 1H), 5.12 (s, 2H), 3.33 (t, *J* = 5.1 Hz, 2H), 2.90 (td, *J* = 6.3, 1.1 Hz, 2H), 2.06 (tt, *J* = 6.2, 5.1 Hz, 2H); ^13^C NMR (151 MHz, DMSO-*d_6_*) δ ppm 148.27, 129.32, 129.26, 129.13, 122.46, 122.22, 121.49, 121.45, 120.95, 120.68, 117.76, 111.17, 111.04, 89.72, 62.07, 55.77, 54.85, 51.42, 49.45, 40.02, 22.48, 22.38; EIMS *m*/*z* 314 (M+); Anal. (C_22_H_22_N_2_): C 84.04, H 7.05, N 8.91, Found: C 83.07, H 7.07, N 8.91.

*3-Chloro-N-((3, 4-dihydroquinolin-1(2H)-yl) methyl) aniline* (**SF8**)

Brown solid; yield 77%; m.p. 219 °C; *R*_f_ = 0.83; IR (KBr) *υ_max_* 3357, 2921, 2835, 2287, 2113, 1795, 1598, 1314, 1197, 989 cm^−1^; ^1^H NMR (600 MHz, DMSO-*d_6_*) δ ppm 7.09–7.03 (m, *J* = 27.7, 6.8 Hz, 3H), 7.01–6.96 (m, 2H), 6.74 (s, 1H), 6.69–6.65 (dd, 1H), 6.54 (d, *J* = 2.9 Hz, 1H), 4.70 (s, 2H), 4.35 (s, 1H), 3.40 (t, 2H), 2.80 (d, *J* = 12.5 Hz, 2H), 2.01–1.97 (m, 2H); ^13^C NMR (151 MHz, DMSO-*d_6_*) δ ppm 147.67, 134.84, 130.66, 127.57, 127.27, 124.27, 121.23, 115.97, 115.36, 114.92, 114.20, 58.76, 48.77, 27.00, 22.40; EIMS *m*/*z* 272.78(M+). Anal. (C_16_H_17_ClN_2_) C 70.45, H 6.28, N 10.27, Found: C 70.49, H 6.23, N 10.28.

*4-Bromo-N-((3, 4-dihydroquinolin-1(2H)-yl) methyl) aniline* (**SF9)**

Green solid; yield 80%; m.p. 200 °C; *R*_f_ = 0.81; IR (ATR) *υ_max_* 3631, 2834, 2287, 1902, 1612, 1313, 1194, 1070, 944 cm^−1^; ^1^H NMR (600 MHz, DMSO-*d_6_*) δ ppm 7.24–7.22 (m, 2H), 7.00 (ddt, *J* = 7.5, 2.0, 0.9 Hz, 1H), 6.88 (td, *J* = 7.5, 2.0 Hz, 1H), 6.67–6.65 (m, 2H), 6.56 (td, *J* = 7.5, 2.0 Hz, 1H), 6.38 (dd, *J* = 7.5, 2.0 Hz, 1H), 5.12 (s, 2H), 3.43 (s, 1H), 3.33 (t, *J* = 5.2 Hz, 2H), 2.90 (td, *J* = 6.1, 1.0 Hz, 2H), 2.06 (tt, *J* = 6.1, 5.2 Hz, 2H); ^13^C NMR (151 MHz, DMSO-*d_6_*) δ ppm 147.42, 144.96, 132.16, 127.52, 127.14, 124.22, 122.56, 119.33, 115.06, 114.31, 111.50, 61.78, 48.79, 26.98, 22.38; EIMS *m*/*z* 316 (M+). Anald. (C_16_H_17_BrN_2_) C 60.58, H 5.40, N 8.83, Found: C 60.54, H 5.42, N 8.86.

*N-((3, 4-dihydroquinolin-1(2H)-yl) methyl)-N-phenylacetamide* (**SF10)**

White crystals; yield 78%; m.p. 220 °C; *R*_f_ = 0.93; IR (ATR) *υ_max_* 3863, 3371, 2931, 2689, 2326, 2110, 1906, 1661, 1598, 1372, 1073, 884 cm^−1^; ^1^H NMR (600 MHz, CDCl_3_) δ ppm 7.49–7.47 (m, 2H), 7.37–7.28 (m, 3H), 7.00 (ddt, *J* = 7.5, 2.0, 1.0 Hz, 1H), 6.88 (td, *J* = 7.5, 2.0 Hz, 1H), 6.56 (td, *J* = 7.5, 2.0 Hz, 1H), 6.38 (dd, *J* = 7.5, 2.0 Hz, 1H), 5.44 (s, 2H), 3.33 (t, *J* = 5.1 Hz, 2H), 2.90 (td, *J* = 6.2, 1.0 Hz, 2H), 2.08–2.04 (m, 2H), 2.05 (s, 3H); ^13^C NMR (151 MHz, CDCl_3_) δ ppm 168.68, 142.24, 137.99, 128.92, 128.22, 128.11, 127.22, 125.55, 124.24, 119.97, 117.44, 49.53, 27.74, 22.57, 22.46; EIMS *m*/*z* 280 (M+). Anald. (C_18_H_20_N_2_0) C 77.11, H 7.19, N 9.99, Found: C 77.13, H 7.19, N 9.96.

*N-((3, 4-dihydroquinolin-1(2H)-yl) methyl)-N-ethylethanamine* (**SF11)**

Off-white solid; yield 81%; m.p. 110 °C; *R*_f_ = 0.86; IR (ATR) *υ_max_* 3851, 3373, 2838, 2235, 2003, 1895, 1610, 1496, 1198, 1093 cm^−1^; ^1^H NMR (400 MHz, CDCl_3_) δ ppm 6.99–6.95 (dd, *J* = 8.2, 1.6 Hz, 1H), 6.93–6.87 (m, 1H), 6.85–6.80 (m, 1H), 6.53 (dt, *J* = 23.3, 7.3, 1.2 Hz, 1H), 3.81 (s, 2H), 3.26–3.18 (m, 2H), 2.69 (td, 2H), 2.60–2.47 (m, 4H), 1.86 (tt, 2H), 0.99–0.92 (m, 6H); ^13^C NMR (101MHz, CDCl_3_) δ ppm 145.93, 129.04, 127.14, 126.88, 122.46, 116.09, 46.98, 27.00, 22.46, 11.88; EIMS *m*/*z* 218 (M+). Anald. (C_14_H_22_N_2_) C 77.01, H 10.16, N 12.83, Found: C 77.04, H 10.13, N 12.83.

*N-((3, 4-dihydroquinolin-1(2H)-yl) methyl)-4-methoxyaniline* (**SF12**)

Pale yellow solid; yield 86%; m.p. 215 °C; *R*_f_ = 0.78; IR (ATR) *υ_max_* 3853, 3393, 2997, 2833, 2323, 2160, 2077, 1887, 1611, 1305, 1154, 1036, 975 cm^−1^; ^1^H NMR (600 MHz, CDCl_3_) δ ppm 6.97–6.87 (m, 4H), 6.73 (dd, *J* = 17.1, 9.0 Hz, 2H), 6.65 (dd, *J* = 8.4, 1.1 Hz, 1H), 6.54 (td, 1H), 4.60 (s, 2H), 4.32 (s, 1H), 3.67 (s, 3H), 3.21 (td, 2H), 2.69 (s, 1H), 1.87–1.84 (m, 2H);^13^C NMR (151 MHz, CDCl_3_) δ ppm 154.07, 144.97, 143.67, 128.92, 127.31, 124.21, 122.18, 120.93, 120.09, 114.47, 55.77, 55.56, 46.97, 26.98, 22.39; EIMS *m*/*z* 269 (M+). Anald. (C_17_H_20_N_2_O) C 76.09, H 7.51, N 10.54, Found: C 76.07, H 7.53, N s10.54.

*N-((3, 4-dihydroquinolin-1(2H)-yl) methyl)-N-(4-hydroxyphenyl) acetamide* (**SF13)**

White solid; yield 84%; m.p. 285 °C; *R*_f_ = 0.91; IR (ATR) *υ_max_* 3249, 2831, 2611, 2063, 1893, 1611, 1454, 1370, 1233, 1019, 833 cm^−1^; ^1^H NMR (400 MHz, DMSO-*d_6_*) δ ppm 9.95 (s, 1H), 7.08–7.06 (m, 2H), 7.00 (ddt, *J* = 7.5, 2.0, 1.0 Hz, 1H), 6.88 (td, *J* = 7.5, 2.0 Hz, 1H), 6.72–6.69 (m, 2H), 6.56 (td, *J* = 7.5, 2.0 Hz, 1H), 6.38 (dd, *J* = 7.5, 2.0 Hz, 1H), 5.44 (s, 2H), 3.33 (t, *J* = 5.2 Hz, 2H), 2.90 (td, *J* = 6.0, 1.0 Hz, 2H), 2.06 (tt, *J* = 6.0, 5.2 Hz, 2H), 2.03 (s, 3H); ^13^C NMR (101 MHz, DMSO-*d*_6,_) δ ppm 168.58, 153.46, 145.77, 130.06, 128.73, 127.15, 127.12, 125.40, 120.95, 115.54, 88.78, 49.27, 27.49, 22.64, 22.39; EIMS *m*/*z* 296 (M+); Anald. (C_18_H_20_N_2_O_2_) C 72.95, H 6.80, N 9.45, Found: C 72.93, H 6.82, N 9.44.

Appendix A report the ^1^H and ^13^C-NMR spectra of all the synthesized compounds.

### 3.2. ADME Predictions

A computational study to predict the ADME (absorption, distribution, metabolism, excretion) properties of the synthesized compounds (**SF1**–**SF13**) was performed using SwissADME software [89] to determine drug-likeness properties. We determined the following parameters: molecular weight (MW), molar refractivity logarithm of the partition coefficient (ilog P_O_/_W_), Alog P, HBA and HBD to forecast Lipinski’s rule of five (RO5).

### 3.3. Bioevaluation

#### 3.3.1. Antioxidant Assay

##### DPPH Assay

The % FRSA (free radical scavenging activity) of the synthetic compounds was determined by using the method of Tai et al. [90] with slight modifications. The antioxidant potential was determined by detecting the capacity of the synthetic compounds to quench the DPPH free radicals. The synthetic compounds were taken in a 96-well plate at three-fold concentrations of 200 µg/mL, 66.6 µg/mL, 22.2 µg/mL and 7.4 µg/mL. DPPH free radicals were added to entire wells to make up the volume up to 200 µL. DMSO and ascorbic acid were taken as negative and positive controls, respectively. The absorbance was taken at 630nm using a microplate reader (ELx800 BioTek). The percentage of scavenging activity was calculated by the given formula:% scavenging activity = (1 − Ab_s_/Ab_c_) × 100

Abs = sample’s absorbance; Abc = control’s absorbance.

##### Determination of Total Antioxidant Capacity (TAC)

The total antioxidant capacity was determined by using the phosphomolybdenum-based method [91]. The method is based on the reduction of Mo (VI) to Mo (V), leading to the formation of a green-colored phosphate–molybdenum complex that gives absorption at 630 nm. In 96-well plates, 20 µL of the sample was added with 180 µL TAC reagents in it and incubated at 95 °C for 90 min in a water bath and cooled at room temperature. The results were calculated as µg AAE/mg.

##### Total Reducing Power (TRP) Determination

The assay is based on the reduction of Fe^+3^ to Fe^+2^. The method described by Khan et al. was used with slight modifications. One hundred microliters of test samples (4 mg/mL DMSO) were taken in an Eppendrof tube and 200 μL of phosphate buffer (0.2 mol/L, pH 6.6) and 250 µL of 1% potassium ferricyanide (K_3_Fe CN)_6_) were added. This was allowed to incubate at 50 °C for 20 min. Later on, 200 μL of 10% trichloroacetic acid (TCA) was added to the mixture. The mixture was centrifuged at 3000 rpm at room temperature for 10 min. After centrifugation, 150 µL of supernatant were transferred to microplate with FeCl_3_ (50 μL, 0.1%) and the absorbance was taken at 630 nm [92]. The results were calculated as µg AAE/mg.

#### 3.3.2. α- Amylase Enzyme Inhibitory Assay

An in vitro α-amylase enzyme inhibitory assay was performed by the standard protocol [93]. Four milligrams per 1 mL of sample was prepared in DMSO. In each well of a 96-well plate, synthetic compounds were added to a final concentration of 200 µg/mL. To this, 15 μL phosphate buffer, 25 mL α-amylase enzyme and 40 μL starch solution were added stepwise. The microplate was incubated at 50 °C for 30 min. Twenty microliters of HCl (1 M) and 90 μL iodine solution (0.254 g I_2_ and 4 g of KI in 1 L of distilled water) were added and the reading was noted at 540 nm. Acarbose and DMSO were taken as positive and negative controls, respectively. The formula used to calculate the percent enzyme inhibition is:% Enzyme inhibition = [(As − An) ÷ (Ab − An] × 100

As = sample’s absorbance; An = negative’s absorbance; Ab = blank’s absorbance

Synthetic compounds with a % enzyme inhibition ≥ 50% were taken at three-fold concentrations of 200 µg/mL, 66.6 µg/mL, 22.2 µg/mL and 7.4 µg/mL and their IC_50_ was calculated.

### 3.3.3. Cytotoxicity Evaluation

#### Brine Shrimp Cytotoxicity Assay

Brine shrimp eggs were hatched in a shallow rectangular plastic dish and filled with artificial seawater, which was prepared with a commercial salt mixture. In a 96-well plate, 150 µL of artificial seawater was added and 10 nauplii were added into this. Test compounds (400, 200, 100 and 50 µg/mL) were added into each well and the final volume was made up by adding the artificial seawater. The well plate was left uncovered under a lamp. The number of surviving shrimps were counted and recorded after 24 h. The test was repeated in triplicate [94]. Using GraphPad Prism, the lethality concentration (LD_50_) was assessed at 95% confidence intervals. The percentage mortality (%M) was also calculated by dividing the number of dead nauplii by the total number and then multiplied by 100.

#### Cytotoxicity Against Raw Macrophages

The cytotoxic potential was assessed by using an MTT 3-(4,5-dimethyl thiazole-2-yl)-2,5- diphenyl tetrazolium bromide assay [95]. Briefly, macrophages were extracted from the peritoneal cavity of albino rats and plated at a density of 1 × 10^6^ per well in a 96-well plate and incubated in a 5% CO_2_ incubator at 37 °C for 24 h. The cells were treated with various concentrations of the test compounds (100 µM, 50 µM, 10 µM and 1 µM) or vehicle alone. The test compounds were first dissolved in DMSO to make a 100 mM stock solution and then further dilutions were made from this stock solution. After 24 h of incubation, 20 µL of MTT (1 mg/mL in normal saline) was added into each well and incubated under the same conditions for 2 h. Mitochondrial succinate dehydrogenase converted MTT in live cells into purple formazan crystals. The formazan crystals were then solubilized in 100 µL DMSO, and the absorbance was measured at 570 nm. The % cell viability was calculated by the following formula;
% cell viability = (abs_sample_ − abs_blank_)/(abs_control_ − abs_blank_) × 100

#### MTT Assay Against Hep-2C Cells

Hep-2C cell line experiments were performed in the Immunology Lab of the National Institute of Health (NIH), Islamabad, Pakistan. Human cervix carcinoma cells Hep-2C (ATCC HB-8065) were cultured in Dulbecco’s modified Eagle’s medium (DMEM), comprising 10% fetal calf serum (10% FCS) and supplemented with 2 mM L-glutamine, 100 U/mL penicillin, 100 µg/mL streptomycin and 1 mM Na pyruvate, at 37 °C in a humidified 5% CO_2_ incubator.

The tetrazolium dye 3-(4, 5-dimethyl thiazolyl-2)-2, 5-diphenyltetrazolium bromide (MTT) was used to assess the cytotoxic potential of the test compounds. In living cells, MTT is reduced into its insoluble purple product formazan, which is measured spectrophotometrically.

Hep-2C cells, in a density of 1 × 10^6^ (> 90% cell viability), were cultured in a 96-well plate and treated with different concentrations of the test compounds for 24, 48 and 72 h. Ten microliters of MTT (1 mg per mL) were added per well followed by incubation for 4 h. Insoluble formazan crystals were dissolved by adding 100 µL of DMSO. Cells were then incubated for another 2 h. Absorbance was measured at 570 nm by a microplate reader [96]. Untreated Hep-2C cells were taken as controls and DMSO as a negative.
% cell viability = (abs_sample_ − abs_blank_)/(abs_control_ − abs_blank)_ × 100

### 3.3.4. Nitric Oxide Assay

#### Isolation of Peritoneal Macrophages and Measurement of Nitrite Production

The anti-inflammatory effect of the synthesized compounds in murine macrophages was evaluated by using the Griess reaction method, described previously [97]. In brief, 1 × 10^6^ were plated in a 96-well plate and incubated in a 5% CO_2_ incubator at 37 °C for 24 h, pre-treated with different concentrations (100 µM, 50 µM, 10 µM and 1 µM) of synthetic compounds for another 2 h and challenged with LPS (1 µg/mL) for an additional 18 h. Equal volumes of Griess reagent (1% sulphanilamide in 50% phosphoric acid and 0.1% naphthyl ethylenediamine dihydrochloride in distilled water and vortexed) and culture medium were mixed and the absorbance was taken at 540 nm. Lipopolysaccharide (LPS) was taken as a blank and piroxicam was taken as a positive control. A *t*-test was applied to determine the significance of the results.

#### In Vivo Anti-Inflammatory Activity

At 7–8 weeks old, male BALB/C albino mice (29–35 g), were purchased from The National Institute of Health (NIH), Islamabad, Pakistan. Five animals were housed per group and placed in a controlled temperature and humidity-controlled room (22 °C and 66 ± 5%, respectively) in a 12 h light–dark cycle and provided with water and food ad libitum. Ethical approval was taken from the bio-ethical committee of Quaid-I-Azam University (Approval No. BEC-FBS-QAU2018-119) and all protocols were conducted following the Ethical Guide to the Code of Practice for the Housing and Care of Animals Act 1986 (ARRIVE (Animal Research: Reporting In Vivo Experiments) guidelines).

The anti-inflammatory activity of the synthesized compounds was evaluated by using the carrageenan-induced rat paw edema assay. To determine the dose response of tetrahydroquinoline derivatives, animals were first given a dose of 0.1, 1 and 10 mg/kg against 100 µL carrageenan (1% solution in normal saline)/paw. Treatment was given 60 min before carrageenan injection. Readings were taken at 4 h post carrageenan injection [98].

The in vivo anti-inflammatory activity was performed following the procedure described by Koksal, et al. [85] in 2017. Paw edema was induced by injecting 100 µL of 1% sterile carrageenan solution into the right hind paw of mice. Synthesized compounds (10 mg/kg), vehicle (saline with 2% DMSO) and drug (acetylsalicylic acid, ASA 10 mg/kg) were administered to the mice orally by gastric intubation 60 min before injecting carrageenan into the right hind paw. The decrease in edema was measured by vernier calipers every 2 h.

### 3.4. Statistical Analysis

The experimental results were expressed as mean ± SEM. Each test was performed in triplicate. The results were statistically analyzed by *t*-tests and ANOVA at a 95% confidence level (*p* < 0.05). GraphPad Prism software was used to calculate the half maximal inhibitory concentration (IC_50_). A value of *p* < 0.05 was chosen as the criterion for statistical significance.

## 4. Conclusions

A series of N-Mannich bases, containing a tetrahydroquinoline nucleus, were synthesized and evaluated biologically for their drug-likeness properties, antioxidant potential, α-amylase enzyme inhibition, cytotoxicity and in vivo anti-inflammatory properties. The results of these assays were found to be encouraging and promising. The compounds exhibited significant scavenging DPPH free radical activity, i.e., **SF8** had the lowest IC_50_ = 29.19 ± 0.29 µg/mL and **SF5** was the most potent in inhibiting α-amylase enzyme, i.e., 97.47%. The compounds were successfully evaluated for their cytotoxic activities on the raw macrophages and anti-tumor activity using the cervix carcinoma cell line (Hep-2C). Compounds **SF1** and **SF7** revealed cytotoxicity to the cervix carcinoma cell line. These results revealed that the cytotoxic potential of new compounds can be a starting point for further surveying them as chemotherapeutic agents in cancer treatment. Moreover, the compounds were further tested on animal models to evaluate their anti-inflammatory behavior. Moderate to potent anti-inflammatory activity of these derivatives containing tetrahydroquinoline was observed, which is comparable with the standard drug, acetylsalicylic acid. As a conclusion, the current study gave us the scope for further work in this area for future developments through derivatization to design more potent compounds. It felt necessary from the results of the current work that there is a need for further advanced studies, at least on the few synthetic compounds which are found to be superior and to produce a rational quantitative structure–activity relationship (QSAR) mapping.

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
