# Peer review of "One-Pot Multicomponent Synthesis and Bioevaluation of Tetrahydroquinoline Derivatives as Potential Antioxidants, α-Amylase Enzyme Inhibitors, Anti-Cancerous and Anti-Inflammatory Agents"

_molecules, 2020, doi:10.3390/molecules25112710_

Round 1
Reviewer 1 Report
After second revision, especially the quality of submitted spectra description and their revision I can recommend the article is ready to publish in Molecules Journal
Author Response
Thank you!
Reviewer 2 Report
Page 2, There is inconsistency for the SF8 structure. In SI Cl is in m-position. Even different structures for SF8 appear in the figuere
Page 7, line 170 "...and d-DMSO were... change to: DMSO-d6
Page 8, line 177 "...pre-HTS era [64]..."
Page 8, line 181 "...is greater than 5 [63,65]. ]..."
Starting on page 24, for spectral data, check against spectra. The signs match the structures, but, there are several errors in the assignment. Some data does not match the spectrum. In these cases, it is preferable to expand regions in the spectrum, to improve signal assignment. I have highlighted some errors found in the manuscript.
13C - NMR data is reported with a decimal.

Author Response
Please see the attachment.

This manuscript is a resubmission of an earlier submission. The following is a list of the peer review reports and author responses from that submission.
Round 1
Reviewer 1 Report
Improve the clarity of schemes.
It is recommended that the authors include mechanistic detail to support the discussion on lines 129 to 133.
Line 33: SF8 structure does not match (manuscript and H-spectrum)
Line 86:"...bezofused...." correct to benzofused
Line 99, 103, 105, 114, 117, 120, 127, 134 and 674 "...mannich..." change to Mannich
Line 140 and 145 "...(deuterated-CDCL3)..." correct to CDCl3
How is the SF 10 compound obtained? Does not match the synthetic route in scheme I.
Reviewer 2 Report
Manuscript describes the synthesis of a series of 13 Mannich bases derivatives of tetrahydroquinoline. Compounds were evaluated in vivo and in vitro for anti-oxidant properties, ability to inhibit alpha-amylase, cytotoxic and anti -inflammatory properties. The subject of the manuscript could be interesting for readers and could fit to the profile of Molecules however manuscript was not prepared carefully, contains several errors and mistakes, not proper description of the obtained compounds (please once more describe the obtained spectra they do not confirm the structures of the obtained compounds, you have a high class equipment so the spectra should look a lot better) and performed tests. In the enclosed text are some main points indicated. Not all mistakes were indicated, with NMR spectra only some examples. The manuscript should undergo major revision and be reconsidered once more for review

Round 2
Reviewer 2 Report
The manuscript has been corrected. However concerning compound SF5. Please let me know which reagents were used to obtain this compound. This compound has two carbons chain. It does not fit to the definition of Mannich base. On the reaction Scheme you have presented that reaction was performed between tetrahydroquinoline, formaldehyde and secondary amine. In the general structure all compounds should have only methylene group between tetrahydroquinoline and amine.Second great problem is that the description of compounds SF2, SF4, SF5, SF6, SF7, SF8, SF9, SF10, SF11, SF12, SF13 spectra, these descriptions do not fit to the provided spectra.
